# *Per1/Per2* Disruption Reduces Testosterone Synthesis and Impairs Fertility in Elderly Male Mice

**DOI:** 10.3390/ijms23137399

**Published:** 2022-07-02

**Authors:** Qinrui Liu, Hu Wang, Hualin Wang, Na Li, Ruyi He, Zhiguo Liu

**Affiliations:** School of Life Science and Technology, Wuhan Polytechnic University, Wuhan 430023, China; m18377610139@163.com (Q.L.); wh592887102@163.com (H.W.); wanghualin313@163.com (H.W.); lina12180@163.com (N.L.)

**Keywords:** circadian clock, *Per1/Per2*, fertility, testosterone synthesis, sperm motility

## Abstract

Circadian rhythm disorders caused by genetic or environmental factors lead to decreased male fertility but the mechanisms are poorly understood. The current study reports that the mechanism of *Per1/Per2* Double knockout (DKO) reduced the reproductive capacity of elderly male mice. The sperm motility and spermatogenic capacity of male DKO mice were weak. Hormone-targeted metabolomics showed reduced plasma levels of free testosterone in DKO male mice compared with WT male mice. Transcriptomic analysis of testicular tissue showed the down-regulation of testosterone synthesis-related enzymes (Cyp11a1, Cyp17a1, Hsd17b3, Hsd3b1, and Star) in the steroid hormone synthesis pathway. Spermatogenesis genes, *Tubd1* and *Pafah1b* were down-regulated, influencing tubulin dynamics and leading to impaired motility. Seleno-compound metabolic loci, *Scly* and *Sephs2*, were up-regulated and *Slc7a11* and *Selenop* were down-regulated. Western-blotting showed that steroid acute regulatory protein (StAR) and p-CREB, PKA and AC1 were reduced in testicular tissue of DKO mice compared to WT. Therefore, *Per1/Per2* disruption reduced testosterone synthesis and sperm motility by affecting the PKA-StAR pathway, leading to decreased fertility.

## 1. Introduction

Mammalian circadian rhythms regulate many physiological, biological, and behavioral processes [1]. The circadian clock encompasses a “master clock” within the hypothalamic suprachiasmatic nucleus (SCN) and “peripheral clocks” within other tissues. The SCN controls circadian rhythms in synchrony with the earth’s rotation and the peripheral biological clock relays the signal to peripheral organs, allowing animals to adapt their feeding, activity, and metabolism to daily environmental changes [2].

At the molecular level, circadian clocks operate through a core transcriptional negative feedback loop plus auxiliary feedback loops. The transcriptional activator, CLOCK/BMAL1, forms heterodimers to initiate the expression of *Period (Per1/2/3)* and *Cryptochrome (Cry1/2)* genes to allow feedback loops to operate. An accumulation of nuclear Per and Cry inhibit *Clock/Bmal1* activity which, in turn, represses Per and Cry, generating oscillations in their expression [3].

Increasing evidence suggests that the deletion of core circadian genes will attenuate circadian rhythms and impair reproductive ability [4]. Male mice with the *ClockΔ19* mutation had reduced fertility, in vitro fertilization rate, and sperm acrosin activity [5]. *Bmal1* gene-deficient mice have disrupted mating behavior accompanied by testosterone levels that are only 70% of that of WT mice [6]. The *Per2* gene modulates the circadian rhythms that influence biological processes and female *Per1/Per2* mutants show premature ovarian failure and impaired ovarian function [7,8]. However, it is not clear how *Per1/Per2* mutations affect the reproductive capacity in male mice. The expression patterns of various clock genes in testicular tissues are not consistent and the relationship with reproductive ability is not clear. For example, Per1 mRNA was only expressed in stage VII–XII of spermatogenesis in secondary spermatocytes [9,10] and Per2 mRNA expression did not show a circadian oscillation in testicular interstitial cells [11]. Clock mRNA is known to be expressed in spermatogonia cells and spermatocytes up to first meiosis [12,13,14], but expression patterns of other clock genes in the testis remain unclear. Testosterone synthesis in testicular tissue shows a diurnal fluctuation, reaching a peak in adult males at 08:00 a.m. and a trough at 7:00 p.m. to 9:00 p.m. [12]. Indeed, sperm concentration and total sperm count were highest in semen collected before 7:30 a.m. [15]. Fluctuations in testosterone synthesis are affected not only by rhythmic genes expressed in the testis but also by circadian rhythms throughout the body, including oscillations of plasma luteinizing hormone (LH) [14,16,17].

Normally, spermatogenesis is well preserved into old age with decreases taking place over decades, so that only 50% of men over 80 years old are completely infertile [18]. However, the impact of long-term abnormal circadian rhythms on the reproductive ability of middle-aged and elderly men is poorly understood. The current study used 15-month-old male mice, considered equivalent to middle or old age, with a double knockout (DKO) of the *Per1/Per2* gene to investigate the effect of abnormal circadian rhythms on the expression of testosterone synthesis genes and corresponding signaling pathways. The influence of microtubule movement and selenium metabolism in DKO male mice is discussed. The aim was to provide theoretical support for male reproductive health in an increasingly aging society.

## 2. Results

### 2.1. Physiological Effects of Per1/Per2 Double Knockout

DKO mice had lower body weights than WT of the same age (*p* < 0.05; Figure 1A), although the mean monthly food intake was not different (Appendix A). Figure 1B shows that the total plasma cholesterol (TC) of DKO mice was significantly lower than that of the WT mice (*p* < 0.05), but free cholesterol (FC) was unchanged. Fasting blood glucose was lower for DKO mice (Figure 1C).

Testes from adult (15-month) DKO and WT mice were observed and weighed. The appearance and oval shape of the testes were similar for the two groups (Figure 1D). However, DKO mice had lower weight testes than WT (*p* < 0.05). Testes: body weight ratios showed no significant difference between the two groups (Figure 1E,F).

DKO mice had sperm of lower motility than WT mice (*p* < 0.05; Figure 1G). Moreover, DKO spermatogenic cells were disordered, spermatogonia and primary spermatocyte layers were thinner and the inner diameter of the convoluted tubules was enlarged with vacuoles compared with WT (Figure 1H,I). We also observed and weighed the testis from adult (nine-month) DKO and WT mice (Appendix A). The results indicated that DKO mice had lower weight testes than WT (*p* < 0.05) (Appendix A). Testes: body weight ratios showed no significant difference between the two groups (Appendix A–C). Fasting blood glucose, sperm motility and testes HE staining showed no significant difference between the two groups. (Appendix A–G).

In summary, the *Per1/Per2* DKO caused reduced body weight, plasma cholesterol, fasting blood glucose, and testis weight, plus lowered sperm motility and spermatogenesis, especially in old mice (15-month). These findings suggest that the fertility of *Per1/Per2* DKO male mice is likely to be severely compromised.

### 2.2. Functional Enrichment Analysis of Differentially Expressed Genes between WT and DKO Mice

Total RNA was sequenced and analyzed. The expression differences were explored by principal component analysis (PCA), which showed distinct clustering images for each group (Appendix A). A total of 532 genes were up-regulated and 425 down-regulated in DKO mice relative to WT. Thresholds of log2FoldChange > 0.1 and *p* value < 0.05 were used to screen the differentially expressed genes (Appendix A) and the results are presented as a heatmap (Appendix A).

GO functional enrichment analyses were performed to assign functions to the dysregulated genes. GO analysis showed that differentially expressed (DE) mRNAs were mainly enriched in microtubule-related pathways, including microtubule-based movement (GO-BP), tubulin binding (GO-MF), and the microtubule (GO-CC; Figure 2). Many circadian molecules (PER2, CLOCK, ARTNL) and cell junction proteins (CTNNA1, CTNNA2) are associated with the microtubule motor protein, dynein (DYNC1H1, DCTN1) [19]. Correspondingly, two microtubule-related genes, *Tubd1* and *Pafah1b* that are involved in spermatogenesis were significantly down-regulated (Figure 2D,F).

### 2.3. Transcriptome Sequencing Revealed Differences in Steroid Hormone Synthesis in Testicular Tissue

KEGG analysis showed that DEmRNAs were mainly enriched in the ribosome and ubiquitin-mediated proteolysis (Figure 3A). The circadian clock has a temporal influence on the translation of a subset of mRNAs involved in ribosome biogenesis by controlling the transcription of translational initiation factors and the clock-dependent activation of signaling pathways. A pathway and pathway interaction network (PPIN) based on the enriched DEmRNAs in each pathway was constructed. Most DEmRNAs were enriched in five pathways, four of which were connected: steroid hormone biosynthesis, cortisol synthesis and secretion, as well as aldosterone synthesis and secretion (Figure 3B). which regulate testosterone synthesis and affect male reproductive ability.

Among those genes which were enriched in steroid hormone synthesis, *Cyp17a1*, *Cyp11a1*, *Ugt1a7c*, *Cyp21a1*, *Hsd17b3*, *Hsd3b1*, *Sult1e1*, *Ugt1a7c* and *Ugt1a6b* were down-regulated in DKO mice (Figure 3C; Appendix A). The genes *Cyp17a1*, *Cyp11a1*, *Cyp21a1,* and *Hsd3b1* were also enriched in the pathway of cortisol synthesis and secretion as well as that of aldosterone synthesis and secretion. Another important gene, *StAR*, which is involved in the same pathways, was also down-regulated (Figure 3D,E). During steroid hormone biosynthesis, the StAR protein transports free cholesterol from the outer mitochondrial membrane to the inner mitochondrial membrane. The CYP11A1 protein catalyzes pregnenolone formation from cholesterol, a rate-limiting step in steroid hormone synthesis [20]. Testosterone may then be synthesized by two pathways, the first, CYP17A1, converts pregnenolone to dehydroepiandrosterone (DHEA), and then Hsd17b produces the testosterone precursor androstenediol; the second, Hsd3b1 converts pregnenolone to progesterone and CYP17A1 produces another testosterone precursor, androstenedione. The two precursors are converted to testosterone by Hsd3b1 and Hsd17b, respectively [21]. Furthermore, pregnenolone participates in early morphogenesis and promotes cell movements by the stabilization of microtubules. Thus, *Per1/Per2* DKO affected steroid hormone synthesis and the plasma testosterone level, which might compromise fertility.

### 2.4. Per1/Per2 Double Knockout Affected the Expression of Genes Related to Steroid Biosynthesis Signaling Pathway

The 9 differential expressed genes involved in the steroid hormone biosynthesis pathway were validated by RT-qPCR (Figure 4A). Expressions of *Cyp11b1*, *Cyp17a1*, *Cyp19a1*, *Cyp21a1*, *Cyp11a1* and *Hsd17b3* were downregulated in the testis of *Per1/Per2* DKO compared with WT (*p* < 0.05; Figure 4A). RT-qPCR results were consistent with RNA-Seq analysis. The genes listed above encode key enzymes involved in steroid hormone biosynthesis, leading to reduced testosterone synthesis and impaired fertility [22]. 

The relative expression of the *StAR* gene, enriched in cortisol and aldosterone synthesis and secretion pathways, was also down-regulated in the *Per1/Per2* DKO testis (*p* < 0.05; Figure 4B). The first step in steroid biosynthesis is the transport of cholesterol from the outer to the inner mitochondrial membrane by steroid acute regulatory protein (StAR) [23]. *Per1/Per2* DKO decreases StAR synthesis, thus affecting steroid hormone biosynthesis. The transcription factor, phosphorylated cAMP response element-binding protein (p-CREB), adenylyl cyclase 1 (AC1), and protein kinase A (PKA) were also downregulated (Figure 4B). All of these proteins are involved in the cAMP-PKA signal transduction pathway. Transcriptome analysis confirmed the corresponding downregulation of the *Creb311* gene in DKO mice (Figure 3D). We propose the following sequence of events: *Per1/Per2* DKO may affect the expression of steroid hormone biosynthesis-related genes, resulting in lower rates of cholesterol transport to the inner mitochondrial membrane, decreased testosterone synthesis, and compromised fertility in DKO mice.

### 2.5. Plasma Testosterone Decreased in Per1/Per2 Double Knockout Male Mice

Plasma hormone levels of nine-month-old and 15-month-old mice were determined by ultra performance liquid chromatography (UPLC)-tandem mass spectrometry (MS/MS). Serum testosterone (T) was lower in DKO mice than in WT at 15 months of age (*p* < 0.05). No significant differences were found in aldosterone (ALD), corticosterone (CORT), 11-deoxycorticosterone (DOC), androstendione (A4), progesterone (P), andluteinizing hormone (LH) (*p* > 0.05; Appendix A; Figure 5 and Appendix A). Decreased testosterone content may be responsible for the lower absolute testicular weight of DKO mice [24]. *Per1/Per2* DKO leads to decreased testosterone and impaired fertility, which is consistent with the above speculation.

### 2.6. Genes Related to Production and Metabolism of Selenoprotein in Per1/Per2 Double Knockout Male Mice

A transcriptome analysis also revealed that the selenocysteine lyase (*Scly*) and selenophosphate synthetase 2 (*Sephs2*) genes were upregulated in DKO mice while the selenoprotein (*Selenop*) gene was downregulated (Figure 6A). Expressions of *Scly 2* in DKO testes were higher than in WT (Figure 6B), suggesting that *Per1/Per2* DKO promoted the cleavage and synthesis of selenocysteine. However, the selenoprotein gene was down-regulated, suggesting that selenocysteine was not further converted into selenoprotein; this may be due to the down-regulation of *Slc7a11* (Figure 6A), and it has previously been shown that disrupting *Slc7a11* reduces selenoprotein expression [25]. Furthermore, the ratio of *Scly* relative expression in testis and brain in DKO mice was lower than in WT mice (Figure 6C). Thus, *Per1/Per2* DKO may affect selenium metabolism in mouse testis, affecting reproductive performance. However, the specific mechanism remains unclear and further research is needed.

## 3. Discussion

Reduced male fertility may be due to several factors, including abnormal circadian rhythms. Timings of hormone release patterns and sperm maturation are critical to male fertility [26]. With the development of society and the aging trend of population, late childbearing has become a realistic demand, therefore, it is necessary to study the effect of abnormal rhythm on fertility in old age. DKO mice showed lower fertility at 15 months of age (about 50 years in human terms [27]) but not at nine months (equivalent to about 35-years in humans [27]) (Appendix A). Compared with the 15-month-old mice, except for the significant difference in plasma free testosterone between the WT and DKO male mice at the age of nine months, there were no significant changes in the transcriptome level, hormone level, or physiological index level (Appendix A). We also analyzed the testicular tissue RNA-seq data of nine month old mice, and the results indicated that there were fewer DEmRNA between DKO and WT mice, especially some important genes observed in 15-month-old mice (Appendix A). Thus, we suggested that decreased fertility resulting from circadian disruption accumulates with age. However, the role of the core clock genes in male fertility remains unclear.

Transcriptomic sequencing analysis of testicular tissue from 15-month-old mice indicated 532 genes up-regulated and 425 genes down-regulated, according to a threshold of FoldChange > 0.1 and *p* value < 0.05. GO analysis revealed that most DEmRNAs were involved in microtubule formation and assembly, includingtwo important gene, *Tubd1* and *Pafah1b*.. Among which, *Tubd1* is involved in Sertoli-Sertoli cell and germ cell-Sertoli cell junction dynamic pathways with the function of testicular Sertoli cells being to convert glucose into lactate to supply energy for spermatogenesis. Moreover, the cycle of the seminiferous epithelium and the spermatogenic wave is initiated by Sertoli cells and maintained by Sertoli-germ cell cooperation [28]. *Pafah1b* depletion significantly impaired spermatogenesis and almost no germ cells were present in some of the seminiferous tubules. Pafah1b normally activates CDC42 in Sertoli cells, which regulates testicular transcription factors via the MAP2K1-MAPK1/3 signaling pathway. [29] The current findings indicate that abnormal rhythms in DKO mice affected tubulin dynamics associated with sperm motility, such as Sertoli cell junction dynamics, and had an impact on spermatogenesis and sperm motility. Thus, decreased Sertoli cell viability due to differential expressions of microtubule-related proteins may account for decreased sperm motility in DKO mice.

A KEGG pathway analysis showed that genes involved in steroid hormone synthesis, cortisol synthesis and secretion, and aldosterone synthesis and secretion had reduced expression in DKO mice. Western blot analysis confirmed that transcription factors in the cAMP-PKA pathway were also reduced. Meanwhile, reduced plasma testosterone levels were seen in DKO mice. In combination, these results showed that the knockout of clock genes affected the testosterone synthesis in mice. It has been reported that *Bmal1*-deficient mice have also decreased testosterone synthesis [6]. However, no significant difference in *Bmal1* expression was found in the DKO mice compared with the WT mice in the present study. Furthermore, there is no direct evidence that the core clock genes, Bmal1 and Per1/Per2, directly impact the testosterone synthesis pathway. Instead, decreased testosterone synthesis may be due to the rhythm disorder itself. Epidemiological studies have suggested that abnormal rhythms are associated with reduced male reproductive ability, including hypogonadism and decreased testosterone levels in male patients with shift work sleep disorder (SWSD) [30]. How might circadian rhythm abnormalities affect testosterone synthesis? Central dysrhythmia may affect reproductive function through the hypothalamic-pituitary-gonad axis [31]. Pituitary LH binds to G-protein coupled receptors in testicular interstitial cells and affects the cAMP-PKA pathway, regulating StAR and the entry of cholesterol into the mitochondrial intima. The cAMP-PKA pathway also controls the synthesis of several steroid hormone synthases, Cyp11a, Cyp17a, Cyp19a, Hsd3β, and Hsd17β, affecting testosterone synthesis and spermatogenesis [32]. The current study found a decreasing trend in plasma LH levels in DKO mice (Figure 5), although levels were not significantly different from WT, and LH receptor (Lhcgr) gene expression in testicular transcriptomes showed no differences. These results suggest a pathway other than AC activation by LH to explain the decrease in testosterone synthesis and plasma-free testosterone levels caused by long-term circadian disturbance.

Testosterone is secreted by Leydig cells and regulates spermatogenesis [33]. Testosterone stimulates a classical pathway, involving the androgen receptor (AR), or multiple non-classical pathways, involving rapid kinase activation, to regulate testicular sperm synthesis [34]. StAR regulates cholesterol transport [23] from the outer mitochondrial membrane (OMM) to the P450scc on the inner mitochondrial membrane (IMM), a process which is regulated by the PKA-CREB pathway [35]. CYP and HSD enzymes, such as Cyp11a, Cyp11b, Cyp17a, Cyp19a, Cyp21a, and Hsd3β, Hsd17β, are involved in testosterone synthesis [21], and the impact of *Per1/Per2* DKO on them is not yet clear. Although TC was significantly lower in DKO mice than in WT and FC, the raw material for steroid synthesis [36,37], showed no significant differences. Therefore, the impact of *Per1/Per2* DKO on testosterone synthesis is more likely to be due to changes in rate-limiting enzymes, such as StAR, rather than to the decreased supply of cholesterol.

SELENOP is a selenium-transporter that maintains antioxidative selenoenzymes in several tissues, including the brain and testis, and plays a pivotal role in selenium-metabolism and antioxidative defense [38]. We found that the ratio of Scly relative expression in testis: brain in DKO mice was lower than in WT mice. However, the body prioritizes selenium supply to the testes, and this change could not have been caused by a dietary selenium deficiency [39]. Selenoprotein P (SELENOP) is a serum glycoprotein required for appropriate selenium distribution in mammals, particularly in supplying selenium to the brain and testes. The SELENOP metabolism is central to mammalian male fertility since it is the sole mechanism for supplying essential selenium to developing spermatozoa [40]. Male mice with a selenoprotein deficiency produce sperm with specific defects in the flagellar structure [41]. *Per1/Per2* DKO may affect selenium metabolism in mouse testis, affecting reproductive performance. Interestingly, changes in genes related to selenium metabolism were observed from transcriptome analysis: *Scly* and *Sephs2* were up-regulated in DKO while *Selenop* and *Slc7a11* were down-regulated relative to WT. Based on these results, we proposed two hypotheses that may lead to decreased fertility in DKO mice: firstly, the reduction of Selenoprotein impairs sperm synthesis and development; secondly, the reduction of Selenoprotein impairs ROS clearance in testis, and the resulting oxidative stress may lead to potential sperm dysfunction [39]. These hypotheses need further research.

Furthermore, previous research also indicated that *Per1/Per2* knockout mice are arrhythmic in con-stant darkness [42], and appearing circadian rhythm weakened or disappeared, including the abnormal rhythm of feeding. Besides, the liver cholesterol metabolism and bile acid synthesis and secretion the food digestion and absorption and the composition intestinal flora of *Per1/Per2* knockout mice are also abnormal, causing the intestinal dysfunction [43,44,45]. Thus, those circadian impairments may also be important factors affecting reproductive capacity in DKO male mice, which may be an all-round influence that deserves further exploration. Finally, the proposed schematics depicting the molecular mechanism of Per1/Per2 in impairs fertility in elderly male mice is present in Figure 7: *Per1/Per2* DKO down-regulated the expression of several key genes and signal regulation molecules involved in testosterone synthesis, resulting in significantly reduced sperm motility, spermatogenic ability and plasma testosterone level. This mechanism may explain the impact of circadian rhythm damage on male reproductive ability. Transcriptome analysis showed that the microtubule activity-related protein genes, *Tubd1* and *Pafah1b1*, were also down-regulated. The deletion of these two genes in mice led to abnormal spermatogenesis [46]. In addition, differential expressions of selenium metabolism-related pathway genes also correlated with normal sperm production. These two pathways may also be involved in decreased fertility in mice, although further experimental verification is required. 

## 4. Materials and Methods

### 4.1. Animal Housing

Male *Per1/Per2* DKO C57BL/6 mice were bred from *Per1*+/−/*Per2*+/− heterozygotes (purchased from the Experimental Animal Center of Soochow University), as described previously [47,48]. The male mice were kept separately in the same environment, with three to four mice in each cage. Control DKO and wild type (WT) mice were divided into two groups (*n* = 20) and housed in ventilated cages at constant temperature (22 ± 1 °C) with moderate humidity (60% ± 10%), fed ad libitum with unlimited access to water on a 12 h light-12 h dark cycle (07:00 a.m. to 7:00 p.m.; the protocol is from SCXK (Hubei) 2016-0011, Wuhan Wanqianjiahe Experimental Animal Company in China) for 15 months. Dietary formulae are shown in Appendix A. All housing protocols and animal experiments were performed following the “Guidelines for The Protection and Use of Experimental Animals”. The study was approved by the Experimental Animal Ethics Committee of Wuhan Polytechnic University (ID: 20190709003). All efforts were made to reduce the number of animals and minimize suffering. All phenotyping procedures were examined for potential refinements. Animal welfare was assessed routinely for all mice involved.

Mice fasted for 12 h were sacrificed by CO_2_ asphyxiation and necks were broken at 7:00 p.m. Orbital blood was collected in heparinized tubes (1.5 mL) and centrifuged at 1000× *g* at 4 °C for 25 min. Plasma was stored at −80 °C for analysis. The testis and epididymis were collected, washed with 0.9% sodium chloride solution, weighed, and isolated testicles were photographed and stored at −80 °C.

### 4.2. Body Weight and Feeding

Growth trajectories were monitored by weekly measurement of body weight to two decimal places. Food intake was measured by calculating the amount of food that was put in the cage (at 7:00 p.m.) minus the weight of the food pellets that were left in the cage at the end of the day (at tomorrow 7:00 a.m.).

### 4.3. Relative Sperm Counts and Motility

Relative sperm counts and motility measurements were made as described previously [49]. Fat was removed from the cauda epididymis, and epididymal tissue was washed with 0.9% sodium chloride solution, transferred to 2 mL Human Tubal Fluid (HTF, Sigma Aldrich, Shanghai, China), cut into pieces and filtered with three layers of periscope paper. Twenty μL HTF was added to a hemocytometer for counting. Dead sperm without motility were counted (X1), and the counting plate was heated at 55 °C for 15 min and the total number of sperm was counted (X2). Motility is given by (X2−X1)/X2. Four counts were made, two each from the left and right caudal epididymis of each mouse and a mean generated from the four counts.

### 4.4. Histological Studies

Testicular tissue was fixed and dehydrated by embedding in paraffin, sectioned with 10% paraformaldehyde for 24 h, dewaxed with xylene, hydrated with gradient ethanol and stained with eosin and hematoxylin (HE). A histological examination was carried out by two independent individuals, and was done in a blinded way. The thickness of the slice was 3 μm, and the interval between each slice was 10 μm. Histopathological changes were observed under a microscope (Nikon-E100, Tokyo, Japan).

### 4.5. LC/MS

Plasma was analyzed by ACQUITY UPLC I-class (Waters, Milford, MA, USA) coupled with mass spectrometry (QTRAP 6500 triple quadruple bar, SCIEX, Shanghai, China) in MRM mode. A Kinetex^®^C18 (2.6 μm, 50 mm × 2.1 mm, Phenomenex, Torrance, CA, USA) column was used with solvent A: 1 mM ammonium acetate and water and solvent B: 1 mM ammonium acetate and methanol. The proportion of solvent B changed from 25–95–25% in 6.5 min. The flow rate was 0.8 mL/min.

### 4.6. RNA Isolation, Library Preparation, and Sequencing

Total RNA was purified from mouse testis (*n* = 9) using Trizol (TAKARA, Kusatsu, Japan). Equal quantities of RNA from three individuals in each group were pooled and mRNA was purified and enriched by oligomer dT coupled magnetic beads (Invitrogen, Carlsbad, CA, USA). RNA was reverse transcribed using random N6 primers and cDNA was synthesized for double-stranded synthesis reaction and sticky ends were generated. Specific primers were used to amplify target DNA sequences and PCR products were thermally denatured to allow circularization of single-stranded DNA with a bridge primer. Single terminal sequencing was performed using DNBSEQ (BGI, Wuhan, China).

### 4.7. RNAseq Data Analysis

Initial sequencing data was analyzed by SOAPnuke software. Reads containing joins, with unknown base N content greater than 5% and of low quality (mass value of <10 bases accounting for more than 20% of total base number) were removed. Filtered reads were compared with the reference sequence. The expression level was standardized with transcripts per million (TPM) and differences in gene expression were analyzed by DESeq2. Differentially expressed genes (DEGs) met the following criteria: log2FoldChange > 0.1; *p*-value < 0.05. GO and KEGG enrichment pathway analyses were performed. Reference genome data were derived from the NCBI database (*Mus musculus* version GCF_000001635.26_GRCm38.p6 https://www.ncbi.nlm.nih.gov, accessed on 12 June 2020). Transcriptome sequencing was commissioned by BGI Medical Laboratory Co., Ltd.

### 4.8. Real-Time PCR Analysis

SYBR Green (SYBR^®^qPCR Supermix, Bio-RAD, Hercules, CA, USA) was used for RT-qPCR assay and Reverse Transcription kit (Takara Biotechnology, Dalian, China) to transcribe 2 μg total RNA into single-stranded cDNA. Primers were designed by Snapgene (Appendix A) and mouse betaactin was used as an endogenous control gene. Each real-time PCR reaction was performed in a final volume of 20 μL containing cDNA reversed transcribed from 0.1 μg mRNA, 1 μL of 10 μM primer and 10 μL of 2 × Universal SYBR Green Supermix (Bio-RAD) in a 96-well plate using a real-time PCR system (CFX96, Bio-rad, Hercules, CA, USA). Amplification conditions were as follows: pre-denaturation at 95 °C for 3 min, denaturation at 95 °C for 15 s, annealing at 60 °C for 20 s, extension at 72 °C for 30 s for a total of 40 cycles. Gene expression levels were normalized to β-actin and fold changes are expressed as 2^−ΔΔCt^ (endogenous reference).

### 4.9. Western Blot Analysis

Total protein was extracted from mouse testis, according to the kit instructions (23252; PILES™Microplate BCA protein Assay Kit, USA). Equal amounts of proteins (40–80 μg) were loaded into wells and separated by 10% sodium dodecyl sulfate-polyacrylamide gel electrophoresis and electrically transferred to a microporous polyvinylidene fluoride membrane. The membrane was blocked with skimmed milk and incubated overnight with anti-StAR (32KD), anti-PKA (41KD) (Proteintech Group, Wuhan, China), anti-AC1(130KD), anti-P-CREB (43KD) (Affinity Biosciences, Cincinnati, OH, USA) or anti-GAPDH (37KD) (Goodhere, Hangzhou, China) antibodies and incubated with a secondary antibody conjugated with horseradish peroxidase for 2 h.

### 4.10. Statistical Analysis

Data were analyzed by GraphPad Prism for Windows, version 7.0 (GraphPad Software, San Diego, CA, USA) and are expressed as mean ± standard deviation (MEAN ± SD). Statistical differences were analyzed by one-way ANOVA (with Tukey’s Test), Mann-Whitney test and chi-square test. A value of *p* < 0.05 was considered statistically significant.

## Figures and Tables

**Figure 1 ijms-23-07399-f001:**
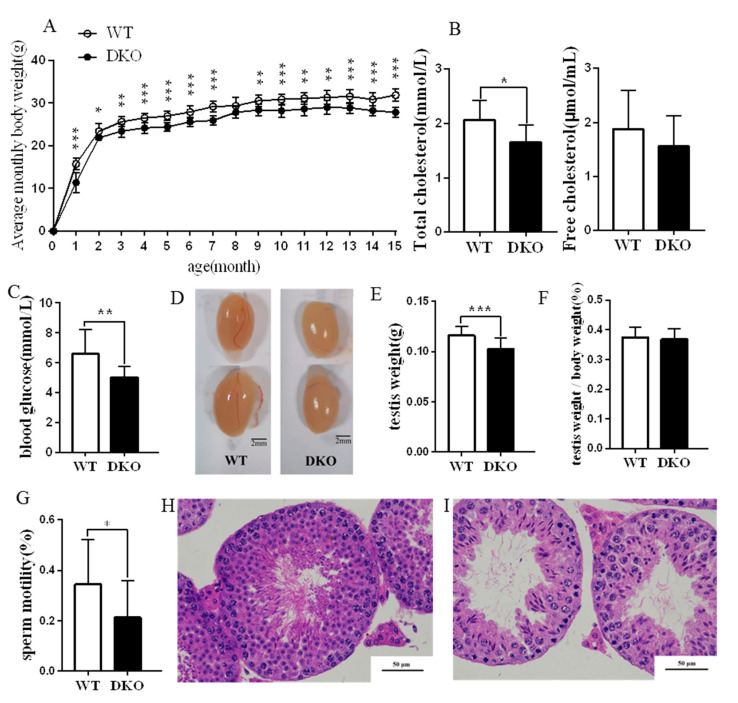
Physiological effect indices and HE staining results of 15-month-age DKO and WT mice. (**A**) Bodyweight growth curve; (**B**) Total cholesterol and free cholesterol; (**C**) Fasting blood glucose levels; (**D**) Testicular appearance (left side at the top, right side at the bottom); (**E**) Testicular weight; (**F**) Ratio of testis weight/body weight; (**G**) Sperm motility; (**H**) HE staining of WT testis; (**I**) HE staining of DKO testis (*n* = 10; * *p* < 0.05; ** *p* < 0.01, *** *p* < 0.001).

**Figure 2 ijms-23-07399-f002:**
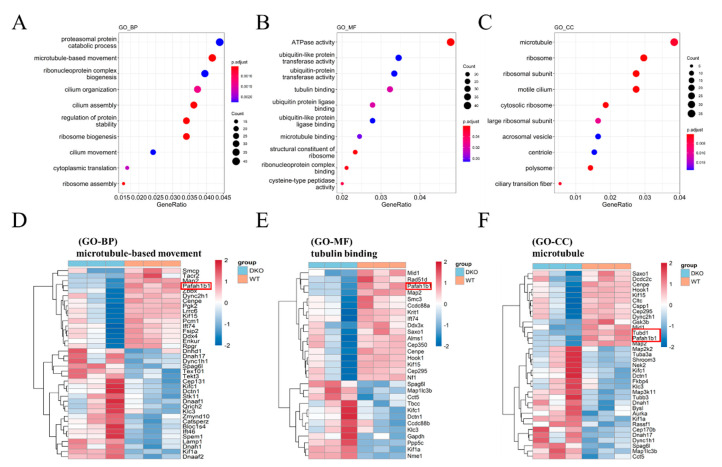
GO analysis of DEmRNA in 15-month-old DKO male mice. (**A**) GO-BP; (**B**) GO-MF; (**C**) GO-CC; Colored dots represent the fold change of gene expression in different pathways. The y-axis represents the gene count. The x-axis gives enrichment analysis terms. Plot colors represent *p*-values and sizes represent gene numbers. (**D**–**F**) Heatmap for differentially expressed mRNAs enriched in microtubule-based movement (GO-BP), tubulin binding (GO-MF), and microtubule (GO-CC). Blue: DKO mice; Red: WT mice. The gene *Pafah1b1* and *Tubd1* were marked in red border.

**Figure 3 ijms-23-07399-f003:**
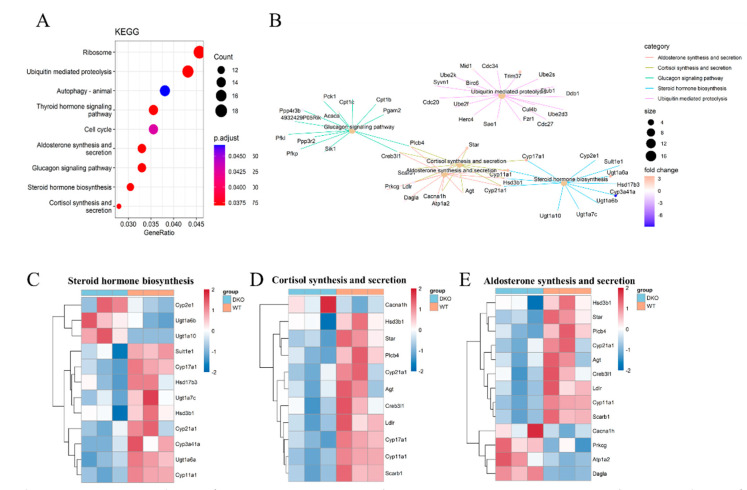
KEGG analysis of DEmRNA in 15-month-age DKO mice. (**A**) KEGG pathway analysis of mRNAs. (**B**) The net plot of KEGG pathways shows the enrichment of DEmRNAs in different pathways. Color dots represent the fold change of gene expression in different pathways. The y-axis represents the gene count. The x-axis gives the enrichment analysis terms. Plot colors represent the *p*-values and sizes represent the gene numbers. (**C**–**E**) Heatmap for DEmRNAs enriched in steroid hormone biosynthesis, cortisol synthesis and secretion, as well as aldosterone synthesis and secretion. Blue: DKO mice; Red: WT mice.

**Figure 4 ijms-23-07399-f004:**
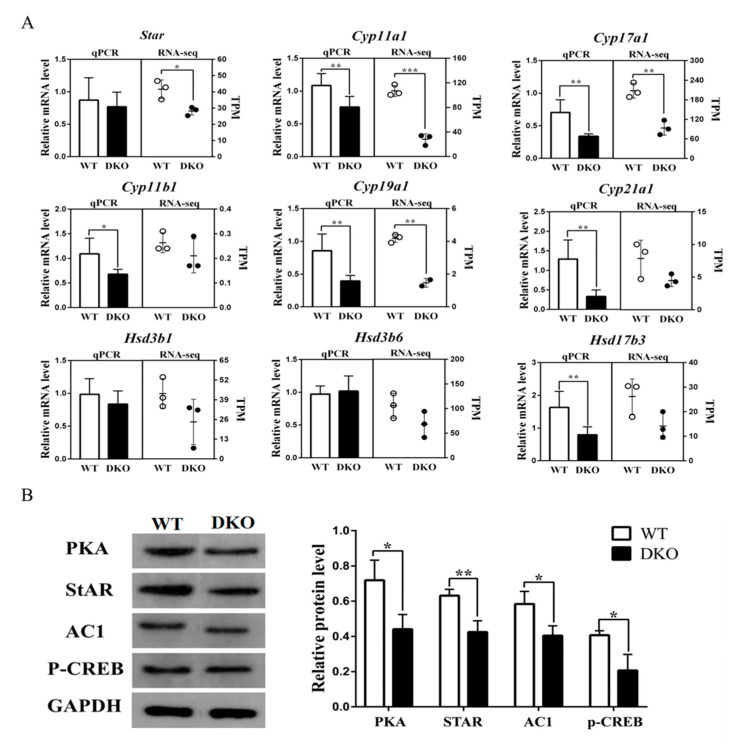
Expression of genes and signaling pathways associated with testicular steroid biosynthesis. (**A**) RT-qPCR results (left side of each image, *n* = 10) and mRNA relative expression in TPM (right side of each image, *n* = 3). (**B**) Protein expression of PKA-StAR pathway-related molecules was detected by Western blotting (*n* = 3). * *p* < 0.05; ** *p* < 0.01.

**Figure 5 ijms-23-07399-f005:**
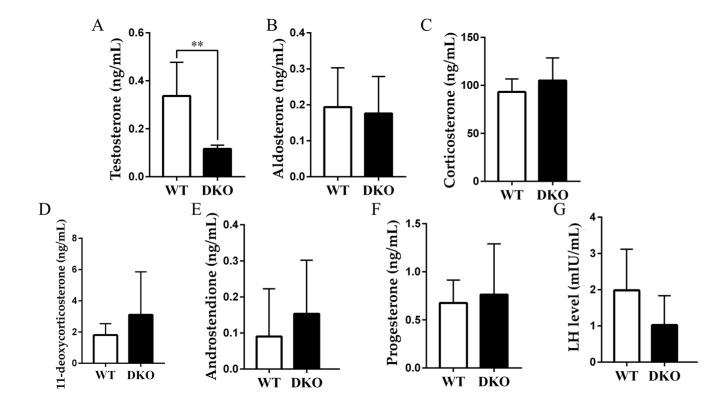
Plasma steroid hormone levels in 15-month-age WT and DKO male mice. (**A**–**G**): T, ALD, CORT, DOC, A4, P, LH (*n* = 8). ** *p* < 0.01.

**Figure 6 ijms-23-07399-f006:**
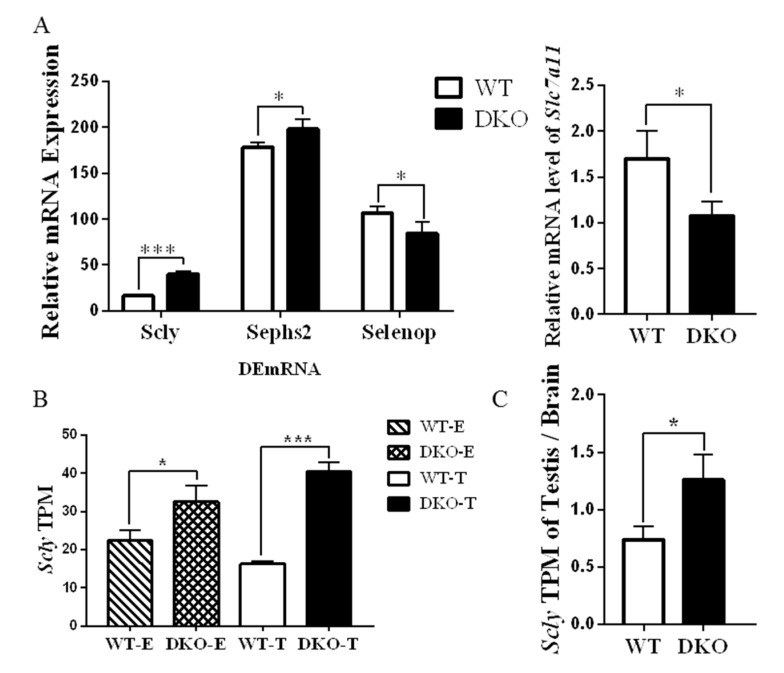
Expression of selenoprotein relative gene. (**A**) mRNA relative expression in TPM of *Scly*, *Sephs2*, *Selenop* and *Slc7a11* genes in DKO and WT testes. (**B**) DKO and WT male transcriptome *Scly* TPM.of testicular and brain (**C**) The ratio of DKO and WT male testicular/brain tissue *Scly* expression. (*n* = 9, * *p* < 0.05, *** *p* < 0.001).

**Figure 7 ijms-23-07399-f007:**
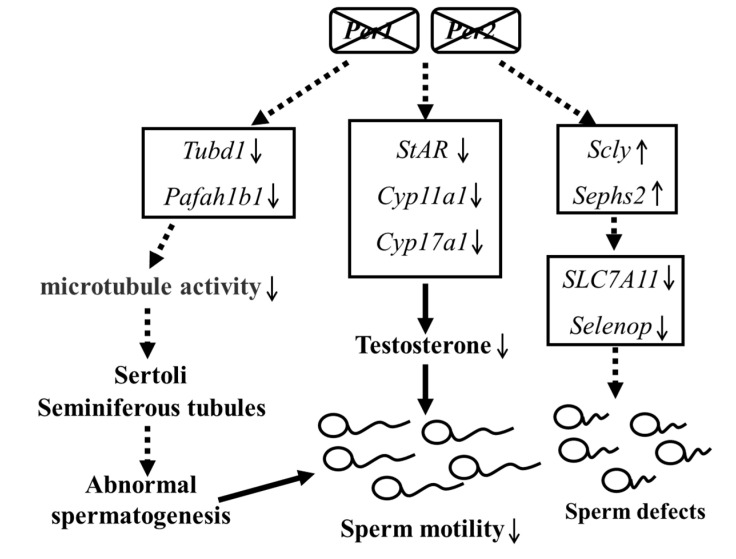
The mechanism by which *Per1/Per2* affect steroid hormone synthesis as well as microtubule activity and selenium metabolism.

## Data Availability

The data presented in this study are available on request from the corresponding author.

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
