# Peer review of "Per1/Per2 Disruption Reduces Testosterone Synthesis and Impairs Fertility in Elderly Male Mice"

_ijms, 2022, doi:10.3390/ijms23137399_

Round 1

Reviewer 1 Report

Liu et al. showed an impaired fertility in elderly PER1/PER2 double KO mice, which is probably due to the down regulation of several genes involved in testosterone synthesis and tubulin related genes as well. The present paper is goal-oriented and well written.

Impaired fertility in clock gene mutant or KO animals are well known and this study with mRNA comparisons between WT and PER1/Per2 double KO add some new insights.

However, there are some suggestions, which would improve the paper and would make it more interesting for the reader.

First, there are passages in results part, which rather belong in the discussion: Page3, line 107 to Page 4, line125 should be shorted in the result part and some of it should be transferred to the discussion. Page 7, line 224 to Page 8, line 236: Same here. Most of it belongs to the discussion.

Secondly, the authors certainly mention that circadian disruption might be involved in reduced male fertility specially in elderly, but they might want to discuss – based on their results- how this might happen. Do you have any data on younger DKO mice besides the body weight? It would be nice to know if there are age differences. If there some data, let the reader know.

Thirdly, PER1/PER2 KO mice are known to be arrhythmic in constant darkness and therefore show circadian impairment in several ways. This should be mentioned and maybe discussed.

At last, Fig. 4: The western blot pictures for AC1 and pCREB do not resemble the reduction shown in the graph. Please check this. Did you also look for CREB?

Page 1 Line 39: space error “…….Per and Cryacts inhibit Clock/Bmal1….” There are more space errors within the paper; please check.

Page 3, Line 91, 94: Please use HE staining instead of H&E staining

Page 4, Line 143: comma error “….and secretion (Fig. 3B). which regulate testosterone…..”

Page 8, line 262: Please use clock genes instead of “rhythm genes” or “rhythm molecules” (Page9, line 266)

Reviewer 2 Report

In my opinion, the manuscript is very interesting and well written. I have only one observation - in the text there are single spelling mistakes - please correct them.

Please see my additional comments to the reviewed manuscript.
Infertility is a great health and epidemiological problem in the modern, especially developed societies. Many men have shift work, what may influence and disturb testosterone levels. What is more couples move the decision about children for later years and some men decide for them in older age. The Authors focused on very important aspects.

I suggest:
-    moderate English and editing corrections (e.g., lines 199-206 - please check leading, line 202 - spaces, line 333 please do not start a sentence with numeral, check typing CO2 lower index)
-    Figure 3B – difficult to read due to very small font – please consider to prepare it e.g., as a separate figure
-    Materials and methods section – please add how many mice were housed in one cage, was the histological examination performed by one or two independents persons and if is was done in a blinded way or not, what was the thickness of the slices (4-5 microm)
-    Please check the references – they should be prepared due the Journal requirements 
